# Low-Cost Data Acquisition System for Automotive Electronic Control Units

**DOI:** 10.3390/s23042319

**Published:** 2023-02-19

**Authors:** João Paulo Bedretchuk, Sergio Arribas García, Thiago Nogiri Igarashi, Rafael Canal, Anderson Wedderhoff Spengler, Giovani Gracioli

**Affiliations:** Software/Hardware Integration Lab, Federal University of Santa Catarina, Florianópolis 88040-900, Brazil

**Keywords:** acquisition system, electronic control unit (ECU), vehicle testing, hardware, controller area network (CAN)

## Abstract

The vehicle testing–validation phase is a crucial and demanding task in the automotive development process for vehicle manufacturers. It ensures the correct operation, safety, and efficiency of the vehicle. To meet this demand, some commercial solutions are available on the market, but they are usually expensive, have few connectivity options, and are PC-dependent. This paper presents an IoT-based intelligent low-cost system for vehicle data acquisition during on-road tests as an alternative solution. The system integrates low-cost acquisition hardware with an IoT server, collecting and transmitting data in near real-time, while artificial intelligence (AI) algorithms process the information and report errors and/or failures to the manufacturing engineers. The proposed solution was compared with other commercial systems in terms of features, performance, and cost. The results indicate that the proposed system delivers similar performance in terms of the data acquisition rate, but at a lower cost (up to 13 times cheaper) and with more advanced features, such as near real-time intelligent data processing and reduced time to find and correct errors or failures in the vehicle.

## 1. Introduction

The global automotive manufacturing sector is one of the biggest industries worldwide and plays an important role in the development of countries with its capital-intensive structure and the volume of employment it creates [1]. In 2020, the automotive market consisted of 85.32 million units and was worth about USD 2.86 trillion in 2021 [2,3]. The industry is expected to grow at a compound annual growth rate (CAGR) of 3.71% from 2020 to 2030, reaching 122.83 million units by 2030 [2].

Automotive manufacturing requires a set of complex project phases during its development process (e.g., requirement specification, system design and simulation, component design, evaluation, vehicle integration, and validation) [4,5]. One of the most important phases is vehicle testing and validation, in which a new vehicle prototype is set to run on real road conditions while a large amount of information (4–6 GB per day) is acquired from its electronic control units (ECUs). The data gathered during the tests provides all of the necessary information to check and validate the powertrain operation, the ECU calibration, emissions, and other parameters.

Due to customer demands for more embedded technology, safety components, strict CO${}_{2}$ emission regulations, and complex powertrain projects, modern vehicles require an extensive test–validation phase to meet all of the requirements, quality, and expectations [6,7,8,9]. Thus, manufacturers have been using equipment and technologies (i.e., expensive testing ECUs and special data acquisition hardware and/or software) that are usually provided by global companies, such as ETAS, Vector, and Bosch. The solutions offered by these companies are usually well suited for the needs of automotive testing, but they have high costs, which help to raise the final cost of the vehicles and limit the usage of such equipment due to budget limitations.

To reduce the development and testing costs of automotive vehicles, many different strategies have been proposed, such as hardware-in-the-loop (HiL), engine-in-the-loop (EiL), and, more recently, X-in-the-loop (XiL) [8,9]. Despite that, on-road testing and validation cannot be replaced because they are important to ensure the entire vehicle’s operation, calibration, safety, and efficiency.

As an alternative solution to the on-road vehicle testing and validation phase, this paper presents a low-cost system for data acquisition during on-road tests. The intelligent acquisition and analysis system for ECUs (IASE) is designed to extract data from the vehicle sensors through the ECU in near real-time, during the on-road testing, process the data, and send them to an Internet of Things (IoT) server with the current vehicle coordinates in SmartData [10] format. This massive amount of data coming from many vehicles at the same time can lead to big data challenges (with the adoption of IoT in manufacturing) and generate industrial big data [11,12].

With the information in the server, the system is capable of processing the ECU data by applying artificial intelligence (AI) to detect anomalies and send all of the necessary information about the vehicle to the manufacturer’s engineers. IASE has a low-cost hardware platform, which is easy to install (just connect a CAN cable from the ECU) and can be used in many vehicle prototypes at the same time as the AI cloud-based system. Therefore, the manufacturer’s testing and validation phase can be optimized, and the vehicle development process can be accelerated (for instance, the engineers can receive a failure report and concentrate the data analysis on the parts that matter). In summary, this paper makes the following contributions:We propose a new low-cost hardware design for data acquisition from ECUs that can be used in on-road vehicle testing and validation processes. The designed hardware presents similar data acquisition rates in comparison with the commercial ones, but it is 13 times cheaper and has additional features, such as near real-time data processing.We integrate the vehicle’s test–validation process into the IoT paradigm, providing near real-time communication via a 4G/LTE network. We also use Bluetooth Low Energy technology to enable communication from the hardware to a smartphone application, providing live information for the driver.We integrate the data collection to cloud-based AI algorithms, enabling near real-time data analysis and quick anomaly detection.We test and evaluate the hardware, obtaining the same results as commercial systems, but with additional features. While commercial systems are PC/tablet-dependent, IASE can acquire and store the data on the IoT server without additional devices.

The rest of this paper is organized as follows. Section 2 describes the background and discusses the main related works. Section 3 shows the proposed system, together with the hardware and firmware descriptions. Section 4 presents an application example, the results, and comparisons between the proposed system and the commercial ones. Finally, Section 5 concludes the paper.

## 2. Communication Protocols and Related Work

This section reviews the communication protocols used to access vehicle ECUs, the current automotive measurement, calibration, diagnosis (MCD) commercial systems, and the works related to the system presented in this article.

### 2.1. CAN, CCP, and XCP

The controller area network (CAN) is a robust, simple, and efficient serial connection created to transfer short messages within a time period (i.e, real-time constraint), making it suitable for communicating with sensors and other time-sensitive devices [13,14]. The protocol defines four different frames: data, remote, error, and overload frames [13,15]. The data field contains up to 8 bytes of data and its content is defined by a higher layer protocol. The current standard (CAN 2.0A and extended CAN 2.0B) supports a transmission rate of 1 Mbps [15].

While the CAN standard defines the physical, data link, and network layer (according to the OSI model), the CAN calibration protocol (CCP) is an application layer protocol [16]. CCP defines the communication between devices using the CAN 2.0B standard and provides means to acquire and calibrate data in ECUs [17].

CCP uses a master/slave architecture in which a master can connect to multiple slaves. As defined by the CAN bus, all messages contain up to 8 bytes of data and are divided into two groups of message objects: the command receive object (CRO) and the data transmission object (DTO). The CRO is used by the master to send a command to a slave and requires a response from the slave in the form of a DTO [17]. To constantly measure variables, data acquisition (DAQ) lists are initialized with object descriptor tables (ODTs) containing the addresses and lengths of variables. The slave then periodically sends DTOs to the master with the information stored in the configured addresses.

The universal measurement and calibration protocol (XCP) is the CCP’s successor and uses a master/slave architecture that supports a single master and multiple slave nodes connected to the same bus. Its transport layer is independent, meaning that XCP can be used with different transport layers, such as CAN, Ethernet, FlexRay, USB, and others [18].

The XCP transfers data in frames composed of a header, a packet, and a tail. Both the header and the tail depend on the transport layer protocol, and the XCP packet is very similar to the CCP frame, containing an identification, timestamp, and data fields. The packet is divided into commands and data. Commands are used to configure, calibrate, and command the slave, and require a response from it. Data are transferred using data transfer objects (DTO) and the DAQ list structure.

### 2.2. Automotive Measurement and Calibration Systems

The three main companies in the sector of automotive measurement, calibration, and diagnosis (MCD) commercial systems are Vector Informatik GmbH, Accurate Technologies, Inc. (ATI), and ETAS GmbH. Table 1 compares features from IASE and these manufacturers. IASE currently supports CCP and XCP protocols over CAN, but other communication protocols (e.g., FlexRay, LIN) can be easily implemented if its bus is supported by the hardware. The JTAG interface is not yet supported but could be implemented in future works.

### 2.3. Related Work

Several systems have been developed to provide communication to vehicle ECUs. For instance, the work presented by [19] shows a measurement and calibration system similar to the commercial ones. Using a personal computer (PC) and a PC CAN interface card, the proposed system aims to communicate with vehicle ECUs through CCP, acquire the vehicle information, and calibrate the ECU based on the link map file of the ECU software. Despite this, the system has no connection to other devices or networks and is entirely dependent on a PC connected to its hardware and the vehicle’s ECU, rising the development cost of the system and making it quite similar to the current commercial solutions.

Other works were developed based on the onboard diagnostics (OBD) system or its newer version, OBD-II [20,21,22,23]. The OBD-II allows connections with the ECU through five different protocols, the CAN protocol being the fastest (500 Kbits/s) [24]. The work presented by [20] shows an in-vehicle data recorder system that acquires data using the CAN bus communications protocol through the SAE J1962 port, the OBD-II. The data are stored in an external memory card and sent to a server using general packet radio services (GPRS), allowing it to be massively distributed for IoT and big data applications. The limitation of the systems developed to work with OBD-II is that only emission-related data can be acquired through this interface (e.g., oxygen sensors, catalytic converters, and emissions-related sensors) [25].

In contrast, direct access to the vehicle information through CAN bus allowed the development of applications used to monitor or control specific subsystems and variables, such as tachographs in cargo vehicles [26], auxiliary loads, and power supplies of electric vehicles [27], as well as the operation speed, engine load, engine torque, and fuel consumption of off-road vehicles [28].

In more advanced applications, communication with vehicular ECUs has provided the data acquisition and control of many variables. With the development of electric and autonomous vehicles, the external control of variables, such as vehicle speed and driver alerts, has been applied with additional sensors and AI algorithms to provide advanced driving assistance [29].

In terms of smart vehicles, GPS localization, the GSM, 4G, 5G networks, and the paradigm of the IoT have been used in works related to smart security systems [30], vehicle speed detection [31], vehicle monitoring systems [32], management of modular batteries for electrical vehicles [33], smart cities [34], and others [35,36]. A combination of the data acquisition through CAN protocol, using OBD-II, the mobile networks, and the paradigm of IoT is proposed by [37] as a smart vehicle monitoring and analysis system, designed to collect real-time data on engine parameters. The purpose of this work includes monitoring and analyzing driving attributes as well as fuel efficiency and vehicle diagnostic data. Despite that, this system cannot be used to acquire all of the necessary data for the vehicular testing and validation process due to the OBD-II limitations as described before.

The work presented by [38] shows the benefits and capabilities of remote data acquisition and edge computing. Using the 5G cellular network, an edge computing platform, machine learning, and other specific techniques and protocols, this work presents a system capable of improving the equipment’s predictive maintenance based on remote data acquisition from multiple sensors and devices.

Although many works have been developed to acquire vehicular data and communicate in different ways with the vehicle’s ECU, none of them presented an integrated system capable of acquiring all of the necessary vehicle data, processing, and sending it through a wireless network, allowing near real-time data evaluation for the test and validation process of vehicle development. Moreover, the use of the IoT concept and AI algorithms to process and store the gathered data are quite new to the vehicle testing and validation process of the automotive industry.

## 3. Intelligent Acquisition and Analysis System for ECUs

The intelligent acquisition and analysis system for ECUs (IASE) aims to be a low-cost alternative to the current commercial equipment available on the market, as discussed in the previous section. IASE is designed to work with the paradigms of the Internet of Things (IoT) and artificial intelligence (AI), differentiating it from the existing solutions. The following sections present the IASE requirement analyses, hardware specifications, experiment configurations, and firmware descriptions.

### 3.1. System Requirements

IASE is motivated by the weaknesses of the current commercial systems, as shown in Table 1. This main motivation has led to the following system requirements:1.Data acquisition through CAN (using CCP or XCP) or JTAG interface.2.Near real-time data upload via 4G following a standardized data format.3.Global position system (GPS) localization.4.Visual or audio status indicators and start/stop commands.5.Obtain the energy to feed the system from the vehicle.6.Processing power to handle the entire data stream.7.Memory capacity for data storage for a full experiment.8.Use a standardized data format for all vehicles to allow the application of machine learning algorithms despite the ECU model.

Based on these requirements, Figure 1 shows a block diagram of the designed IASE data flow, where the hardware is the main component and communicates with all of the other parts of the system. The setup interface represents the initial configurations, where the manufacturer team should define the information (i.e., variables) that needs to be extracted from the vehicle ECU during the tests and its acquisition period. After receiving these details, the IASE Hardware starts to acquire data from the vehicle ECU. Next, the driver interface allows the driver to start the test acquisition and stop it when the experiment (a test) is done. While the acquisition is being made, the IASE hardware processes the data and sends it to the IoT server using a specific data format: SmartData with standardized values based on the International System of Units (SI) (requirement 8). Finally, the data are processed by machine learning algorithms and sent to the manufacturer’s specialists.

### 3.2. Hardware Description

The IASE hardware can be divided into seven blocks that represent the main parts of its architecture, as shown in Figure 2. The connection between the vehicle ECU and the IASE hardware is done either by CCP or XCP. The physical connection between the ECU and the CAN interface is made through DB9 connectors.

To provide a 4G network connection and GPS localization (requirements 2 and 3), the hardware has a Mini PCIe LTE category 4, the Quectel’s module EC25-AU. This module can deliver up to 150 Mbps download and 50 Mbps upload rates, supporting LTE, WCDMA, and GNSS technology [39]. Moreover, a GPS and a Wideband 4G LTE antenna are used to ensure signal quality, while a SIM card of a local mobile phone operator is used to allow 4G access. The connection with the module can be made through its UART and USB ports using serial protocols.

The IASE driver interface is provided by a smartphone application that communicates through Bluetooth Low Energy with the main hardware, showing some status indicators and enabling commands to start and stop the acquisition process (requirement 4).

The Espressif Systems module ESP32-WROOM-32D is used in the IASE hardware as the Bluetooth module. This device is a Wi-Fi+BT+BLE (Wi-Fi, Bluetooth, and Bluetooth Low Energy) MCU module that has two CPU cores with clock frequency adjustable up to 240 MHz. In addition, the ESP32-WROOM-32D has many hardware interfaces, including UART and SPI ports. The module manages all of the BLE configurations, allowing the smartphone application to connect and communicate with the hardware in real-time (requirement 4).

Apart from the smartphone application, IASE uses a LED display to indicate the system status and a buzzer to warn the driver about some status change or eventual error (requirement 4). The control of these elements is made through the processor’s GPIO pins.

To provide each part of the system with appropriate voltage levels (requirement 5), a set of circuits was designed to regulate the external voltage provided by the vehicle battery. These circuits are designed to source, with the correct voltage level, the processor block, the 4G/GPS modem, and the Bluetooth module.

The processor unit is the main part of the hardware, connecting all of the modules, peripherals, and the vehicle ECU itself (requirement 6). IASE uses the FZ3 deep learning computing card, produced by Shenzhen Myir Technology. This card has a system-on-a-chip (SoC) composed of a programmable logic part (Kintex Ultrascale+ FPGA) and a processing subsystem (ARM quad-core Cortex-A53 and dual-core Cortex-R5). The card works with a 4 GB DDR4 SDRAM and has many peripheral interfaces and resources, such as SD/MMC interface, USB 2.0 and 3.0, Ethernet interface, QSPI, and UART ports.

The communication with the vehicle ECU is performed through the CAN interface peripheral of the SoC. Using the CCP/XCP (requirement 1), the processor sends all of the experiment specifications to the vehicle’s ECU setting, where data and the related acquisition frequencies are required. After that, the ECU starts to send the required data to the hardware periodically.

Using the USB port of the FZ3 card, the processor is connected to the EC25-AU module with a baud rate of 115,200 bits/s, allowing the acquisition of the GPS data and the upload of the vehicle data processed to the IoT server. In parallel, the communication with the Bluetooth module happens through the serial ports of the FZ3 card and ESP32 with a baud rate of 115,200 bits/s.

A 16 GB MicroSD card connected to the FZ3 card through the SD interface is used to boot the SoC with its operating system (Linux). Moreover, the MicroSD card is also used to store the collected vehicle data while it is not uploaded to the server (requirement 7).

All of the devices, modules, and electronic components of the IASE hardware are assembled in a printed circuit board (PCB) to provide the necessary physical connections for the system. Figure 3 presents a top view of the PCB with the FZ3 card connected to it.

### 3.3. Experiment Specification and Configuration

When an ECU is configured and installed in a vehicle, an ASAM MCD-2 MC Language (A2L) file is generated. This file contains all of the necessary information regarding the ECU’s internal calibration and measurement variables. We use the A2L file as input to specify and configure an experiment.

In IASE, the A2L file is parsed into a JavaScript Object Notation (JSON) file beforehand and passed to a configuration software which reads the JSON file and allows the automotive engineer to select the variables to be monitored and their sampling frequency through a graphical user interface (GUI). The selected variables are included in another JSON file, called experiment JSON, which is then uploaded into the IASE hardware platform. The software also considers the bandwidth limitation imposed by the CAN network and the ECU itself, and by reading a raster (measurement frequency) property called "length", it knows how many ODTs each raster supports. Therefore, the bandwidth is never exceeded, and it is possible to calculate each raster usage.

The configuration software also includes conversion information in the experiment JSON. This information is necessary since the values obtained from the ECU are commonly not physical values. The software then converts the obtained data to appropriate units of the International System of Units (SI). Thus, in the server, intelligent algorithms can be used, despite the unit an ECU uses (i.e, we have standardization in terms of data units).

The software uses the converted data and transforms it into SmartData, including the necessary metadata which makes it self-contained regarding spatial location, timing, semantics, and truthfulness. The SmartData includes value, unit, the position where it was generated, the timestamp of the moment it was produced, and its validity [10]. The software stores the SmartData in SmartData-Series, which is a group of elements generated in the same device with the same unit and with the same acquisition frequency. The number of data in the series is not fixed and is adjusted according to the requirements and network limitations.

### 3.4. Firmware Description

IASE firmware controls all of the features performed in the hardware. It is programmed using C++ and runs on the Linux kernel 5.4.0-Xilinx-v2020.1. The code controls the interface with the ECU, communication server, and UI. A state machine manages the logic of the firmware depending on the status of the system. Figure 4 depicts the UML state diagram of the main state machine of the IASE firmware. Rectangles represent the states, while the arrows with the event names represent the state’s transitions. After starting the firmware, all classes (i.e., DataController, ECUReader, etc.) are initialized and checked. Then, the firmware remains in IDLE mode reading and writing into the ECU until the user requests a new cycle or finalizes the process.

## 4. Evaluation

To evaluate the proposed system, we used a vehicle, Sandero 2019 1.6, provided by the company Renault S.A. with an ECU capable of communicating via CCP protocol. Based on the available ECU specifications, we configured the IASE hardware to acquire the maximum number of variables allowed by the CCP protocol (258 variables). Then, we measured the time to acquire, process, and send the ECU data, and the CPU utilization, memory, and power consumption.

To demonstrate how the system can be used with AI technologies, we present a machine learning application for data classification related to the vehicle’s fuel consumption. We finish the evaluation by comparing IASE with one commercial system in terms of features, performances, and costs.

### 4.1. Performance Tests and Results

Using the software developed to set up the experiments for IASE, a test experiment with the maximum number of variables and minimum acquisition period allowed was created. The limitations of variables number and periods are related to the maximum bandwidth imposed by the CAN network and the vehicle’s ECU. The created experiment was configured according to Table 2, resulting in a total of 258 variables (equivalent to 477 bytes of variables information per cycle). While 166 variables were acquired with a fixed period, 92 variables were acquired at each cycle of the engine’s cylinders.

The implementation of the experiment in IASE was successful, showing that the system can acquire the entire dataset using CCP. Considering that the limitations of the data that can be transferred through the CAN interface are related to the CAN network and the vehicle ECU, the IASE acquisition capacity reached the maximum possible value of the acquisition rate for this configuration.

In the same way, the system showed to be capable of processing all of the data collected and sending it through the 4G modem to the IoT server. Figure 5 shows the cumulative number of bytes (y-axis) transferred to the server over time (x-axis). The measured rate of bytes over time includes all of the processes of acquiring, processing, and sending the data provided by the ECU. The measured average rate was 330 kB/s and its related rate of variables values was 17,727 elements/s. At this rate, the system is capable of sending all of the variables acquired in near real-time with a small and stable delay of 1 to 3 min, depending on the experiment setup.

While the acquiring and processing times are generally stable, the sending time is dependent on the 4G signal quality and the network speed. Figure 6 shows the average data transfer speed in kB per second over time for the tests made with the presented experiment. The average measured transfer rate was 726 kB/s and its related number of variables values was 23,131 elements/s. This result shows that the 4G module, when the signal quality is good, is capable of sending all of the collected data for a full experiment running in the system.

Figure 7 and Figure 8 depict the CPU utilization and memory consumption of the IASE firmware, respectively. The results include the consumption and utilization of the IASE operating system and firmware at the same time. We measured both metrics through a script to constantly read the record of these statistics in Linux. The CPU utilization in the y-axis of Figure 7 oscillates around 5% changing to values close to 10% and having peaks of around 28% when the data are prepared and sent to the server. In Figure 8, the y-axis represents the memory allocated to the IASE firmware in GBs. The value increments from 1 GB to 1.6 GB when the internal buffers are filled with the data received from the ECU, when the process is stable the value remains around 1.6 GB until the experiment is finished. The initial value is the result of memory reservation done when some of the internal buffers of the firmware are constructed.

The system power consumption was measured using a 0.1 ohms resistance in series with the IASE hardware and an oscilloscope connected to both sides of the resistor. The voltage drop over the resistor was measured during the execution of IASE and the power consumption was calculated afterward using these measures. Figure 9 shows the power consumption of the system over time, presenting an average power consumption of around 10 W. During the system measurement, it was possible to notice that the power consumption peaks occurred when the 4G network was used to send the acquired data through the EC25-AU module, showing that this device has an important share in the power consumption of the hardware.

The data stored in the IoT server can be downloaded through software applications or visualized in web applications, such as Grafana [40]. For instance, Figure 10 shows an example of the data acquired from the vehicle using IASE and displayed in the Grafana dashboard. The variable displayed is the vehicle engine speed and the variable unit is Hertz.

### 4.2. AI Application

As explained, the vehicle’s desired variables can be selected for the most diverse applications. Once the JSON experiment is configured and loaded into the vehicle’s ECU through the IASE hardware platform, the variables are measured and sent to the IoT server for manipulation and storage. This section presents an analysis of the fuel consumption profile that the car driver has on short trips (up to 15 min) to classify it as economic or non-economic, driving the Renault Sandero and monitoring the following variables:Engine speed.Vehicle speed.Position of the accelerator, clutch, and brake pedals.Fuel consumption.Fuel/air mixture.Intake air temperature.Intake manifold temperature.Engine coolant temperature.Total vehicle distance.Throttle valve position.Tooth-to-tooth engine speed.

Vehicle consumption is relative and varies between different car models, so the ECU does not contain a consumption classification variable that defines what is high consumption and low consumption. This information cannot be acquired directly from the ECU to be used as a training parameter or for validating results.

For this reason, the K-means clustering algorithm available in the Python scikit-learn [41] library was used, it is an unsupervised learning algorithm (learns patterns from unlabeled data) that evaluates and clusters the data according to its characteristics. K = 2 was defined as the number of clusters, so it computes K-means clustering, computes cluster centers, and predicts the cluster index for each data sample [42]. The algorithm was trained with real data collected in the experiments and configured to classify the obtained data into two groups: economic (0) and non-economic (1). Therefore, we drove the car for short periods (15 min), varying the driving patterns to simulate economic and non-economic drivers (we changed the direction, speed, and rotation). The variables defined above were collected and sent to the IoT server, which then ran the K-means algorithm.

Figure 11 shows the results of the samples obtained in the driving tests, illustrated in a graph of the vehicle speed (km/h) vs. engine speed (rpm). The points highlighted in red are not economic and those in blue are the economic ones; the car was forced at the end of the route.

Figure 12 illustrates the engine speed graph (in blue) and the cluster graph (in red), where the high consumption points are strongly related to the high rotation of the car’s engine, showing the importance of the variables about the consumption profile.

To validate the clustering, first, we consider the behavior of the driver, knowing that at the end of the experiment, the vehicle was driven more aggressively, expecting a higher consumption. Second, using data from two experiments carried out, lasting 15 min each, we monitored the frequency of vehicle consumption (Figure 13).

The average consumption was around 4.77 L/h, with a median of 4.05 L/h. A consumption label was created using the average as a threshold; consumption higher than this indicates that the car is being forced and consuming more. A confusion matrix was used to compare the expected labels and those obtained by the algorithm; the results showed accuracy and precision close to 75%.

Regarding the general performance of the model in all classes, the accuracy value was considered sufficient to exemplify the use of the IASE, as both classes are equally important for the application. In terms of comparison, the authors of [43] presented an accuracy value of 79.31% in the clustering stage of their work, allowing the creation of other analyses.

The presented binary classification problem is an example of how the IASE hardware can enable AI applications to provide better information to manufacturers, validating the system architecture (from data acquisition to data analysis). The work presented by [44] is another example of the hardware’s use, where the necessary data were acquired through the IASE system. In future works, we will focus on other works that use IASE hardware and the application of AI in data analyses.

### 4.3. Comparison with Commercial Systems

In this section, we compare IASE with a commercial system, the ETAS ES581 CAN Bus interface USB module [45] together with the ETAS INCA software [46], in terms of features, performance, and costs.

#### 4.3.1. Features

With regard to the acquisition methods, both IASE and ETAS systems use the same method to communicate with the ECU, set the experiment variables, and receive the data requested. Moreover, the protocol and communication speed are the same, so the systems do not differ in this specific feature.

One difference between the two systems is that while INCA is centralized, using one single device to acquire, process, store, display the data collected, and interact with the driver, IASE is designed to interact with other devices and platforms. The acquisition and processing steps are made by the IASE hardware, the data are stored in the IoT server, a dashboard displays the acquired values on a webpage to provide near real-time visualization, and the cloud server can apply AI algorithms as soon as new data arrive, generating alerts to the engineers in case an anomaly is found. In a case like this, the problem can be analyzed and the on-road test can be stopped, or restarted, or a new experiment can be configured to provide more information related to the specific condition. This feature allows the manufacturer to improve the vehicle’s testing phase efficiency.

In contrast, the solution provided by ETAS can only upload the acquired data when the experiment is done. In this case, if an experiment has errors, the driver procedure is incorrect, the vehicle presents some critical fault, or some variables are out of range and the experiment needs to be restarted, reconfigured, or stopped, it can be verified only after the driver finishes the on-road test and the experiment file is uploaded to the server and processed, which is time-consuming and expensive.

Another difference between the two systems is the driver interface. While ETAS does not have a specific interface to communicate the experiment status to the driver, IASE uses a smartphone and a LED display with visual information and a buzzer to indicate the system status and/or possible errors.

In addition to the acquisition process, both INCA and IASE can act in the ECU calibration, setting many adjustable parameters through the CCP protocol. This feature allows the system to be used in earlier phases of vehicle development, setting ECU parameters to improve efficiency.

Another important feature of vehicle testing systems is the installation process, which needs to be easy enough to not raise the setup time or demand complex tasks. In this way, IASE is similar to ETAS, working with DB9 connectors to access the ECU CAN and one power source plug that can be placed in any 12 V connector in the vehicle.

#### 4.3.2. Performance

The IASE processing and acquisition speed is limited by the communication between the ECU and the hardware. The CCP used to implement this communication allows baud rates of up to 1 Mbit/s (ISO 11898-2). As a consequence of this limitation, the number of variables that can be acquired from the ECU is also restricted. Therefore, the range of variables that can be set up in an experiment varies accordingly to their size and acquisition frequency. Thus, in terms of acquisition speed, IASE and ETAS present the same performance, because both communicate with the ECU using the same methods and have the same communication speed limitations.

Considering that both IASE and ETAS receive data from the ECU through digital protocols, the possible divergence in accuracy can only be assigned to possible divergences in the conversion methods or errors in data transmission. Intending to check the correctness of data acquisition and processing, the resulting value of many vehicle variables was compared. In this criteria, no differences were found between the two systems.

IASE uses a specific data format, as discussed above, named SmartData. By converting ECU raw data to SI-based values, it is possible to apply the same AI algorithms in different ECU models. INCA has no such feature, which requires the engineers to perform unit and/or data conversions during their analyses.

#### 4.3.3. Costs

In terms of costs, ETAS, similar to most commercial systems, presents expensive hardware parts and an annual license for INCA. In contrast, our system does not require any license and presents hardware parts with lower costs. Table 3 shows the cost comparison between INCA and IASE.

Considering just hardware parts, the IASE solution is about nine times cheaper than INCA, proving its viability. Including the annual license required to use the INCA software, the cost difference increases by almost 13 times.

## 5. Conclusions and Recommendations Conclusion

This paper presented an alternative low-cost solution for vehicle data acquisition during on-road tests, showing its feasibility in terms of costs and the benefits of new features that are capable of improving the vehicle’s testing and validation process. The developed hardware allows the manufacturer to acquire vehicle data remotely, which can improve the efficiency of the vehicle’s test–validation phase. At the same time, the hardware is easy to install and use, allowing it to be used in multiple test vehicles at the same time without limitations. Different AI algorithms can be applied to the gathered data to give more detailed data about the behaviors of the vehicle and driver, allowing for a more in-depth analysis of the vehicle’s performance. Another conclusion is that, together with IoT technology, the data collected from many vehicles can generate industrial big data, which can allow the development of new analyses based on big data techniques.

In comparison with a similar solution provided by ETAS, the IASE is up to 13 times cheaper and has the same performance results. System measurement results proved that the hardware is capable of processing and updating all acquired data to the server at a rate of 330 kB/s. The main additional features of the hardware include 4G, GPS, and Bluetooth connectivity, allowing the system to work with IoT technology and interact with a smartphone application.

The system’s capability of acquiring, processing, and sending the data in near real-time showed to be a major difference in comparison to commercial solutions that usually allow the data analysis only after the whole testing process is finished. This feature can be an important asset if something unexpected happened during on-road tests, allowing the manufacturer specialists to analyze the problem and take immediate actions if necessary, restarting, stopping, or reconfiguring the experiment.

In summary, IASE presented a similar performance in comparison with commercial systems, new features that can improve the effectiveness of the on-road test, and a much more attractive hardware cost than the average commercial equipment.

This initial hardware development still has many possible improvements, allowing future work. In terms of hardware, the system can be optimized, reducing hardware costs or improving connectivity characteristics. Concerning the data upload, a new modem with a 5G cellular network could speed up data transmission and a Wi-Fi network could be set up to upload the data at the end of the tests when the cellular network is not able to do it for any reason. The system can also be enhanced to work with the JTAG interface, increasing the acquisition rate and the number of variables that can be acquired at each new test.

Despite the main objective of the developed hardware, the acquisition system can be easily adapted for different applications. Based on the processing power and connectivity of the hardware, some modifications to the acquisition system could collect data directly from sensors or other vehicle communication buses. Finally, with major modifications, applications in other industrial scenarios would also be feasible.

## Figures and Tables

**Figure 1 sensors-23-02319-f001:**
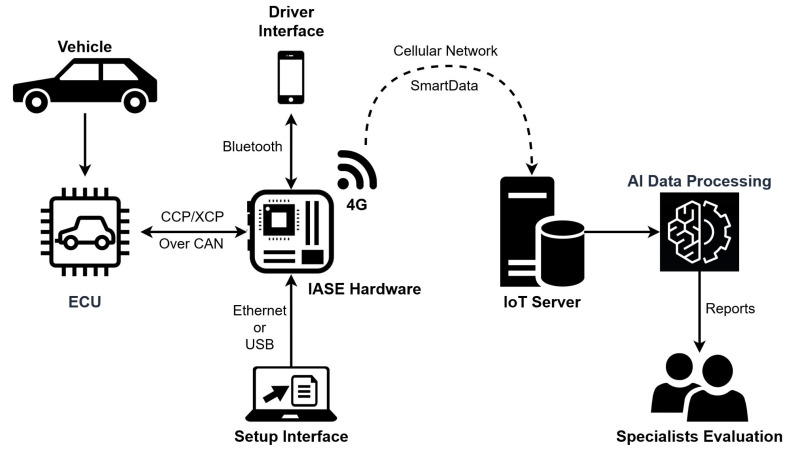
IASE Data Flow Overview.

**Figure 2 sensors-23-02319-f002:**
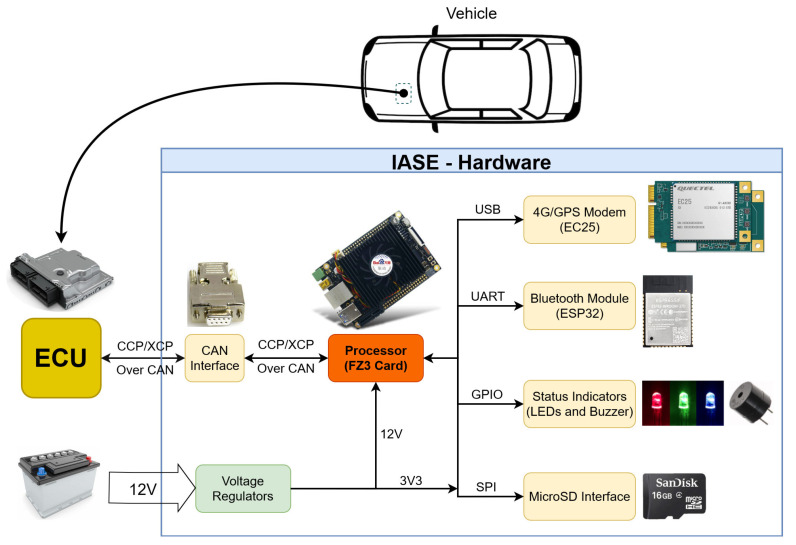
Block Diagram of IASE Hardware.

**Figure 3 sensors-23-02319-f003:**
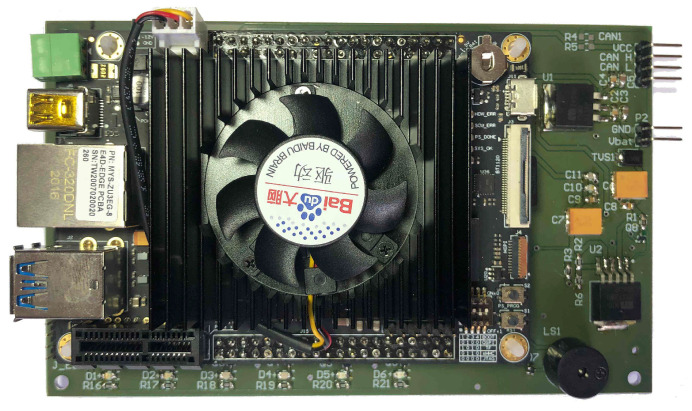
IASE hardware and FZ3 card top view.

**Figure 4 sensors-23-02319-f004:**
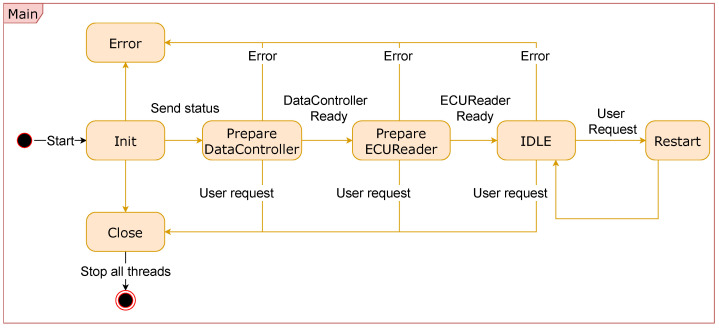
IASE main state machine.

**Figure 5 sensors-23-02319-f005:**
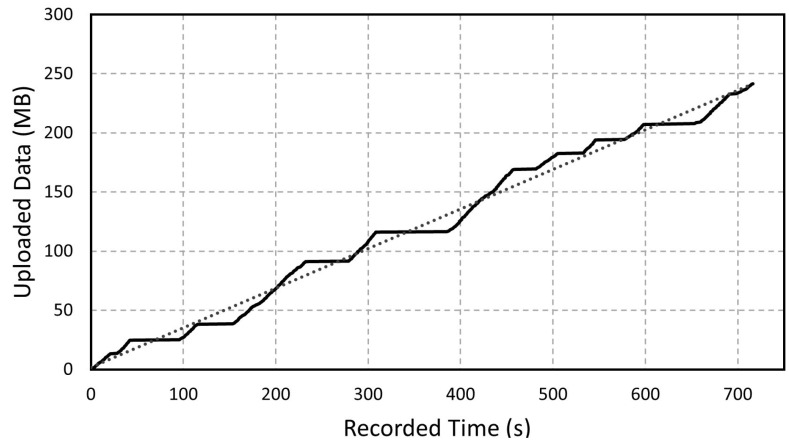
The number of bytes acquired, processed, and sent to the IoT server over time. The solid line represents the measured values and the dotted line represents the average value.

**Figure 6 sensors-23-02319-f006:**
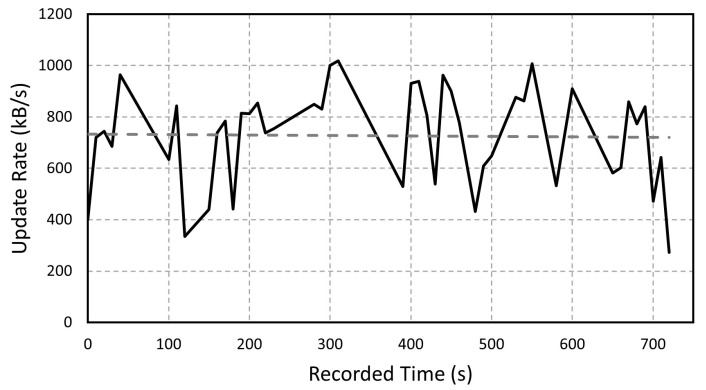
Average data transfer speed over time. The solid line represents the measured values and the dashed line represents the average value.

**Figure 7 sensors-23-02319-f007:**
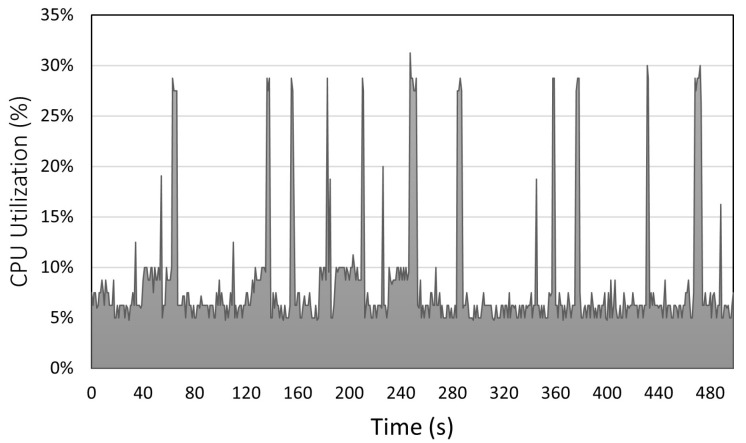
IASE CPU utilization over time.

**Figure 8 sensors-23-02319-f008:**
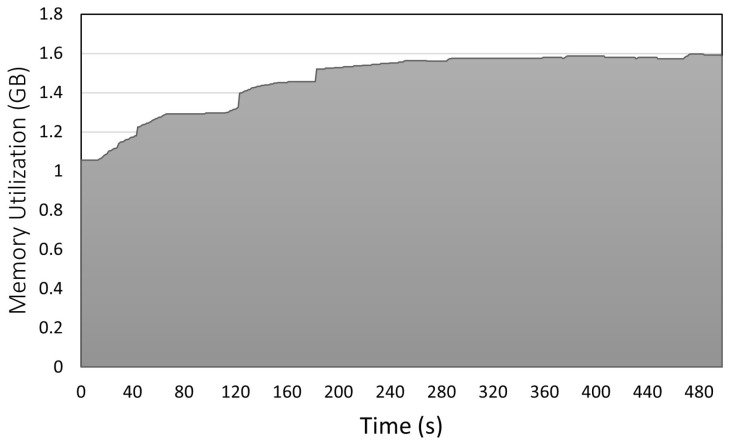
IASE RAM utilization over time.

**Figure 9 sensors-23-02319-f009:**
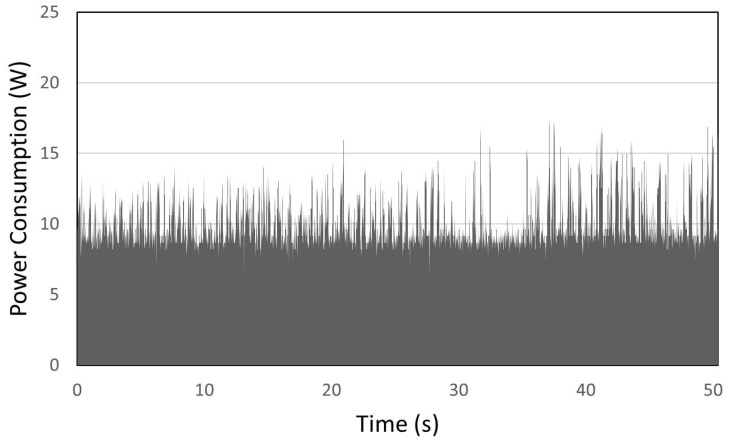
IASE Power consumption over time.

**Figure 10 sensors-23-02319-f010:**
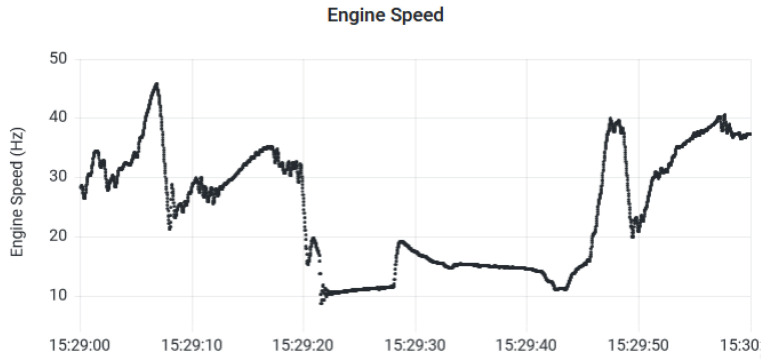
Grafana dashboard vehicle engine speed.

**Figure 11 sensors-23-02319-f011:**
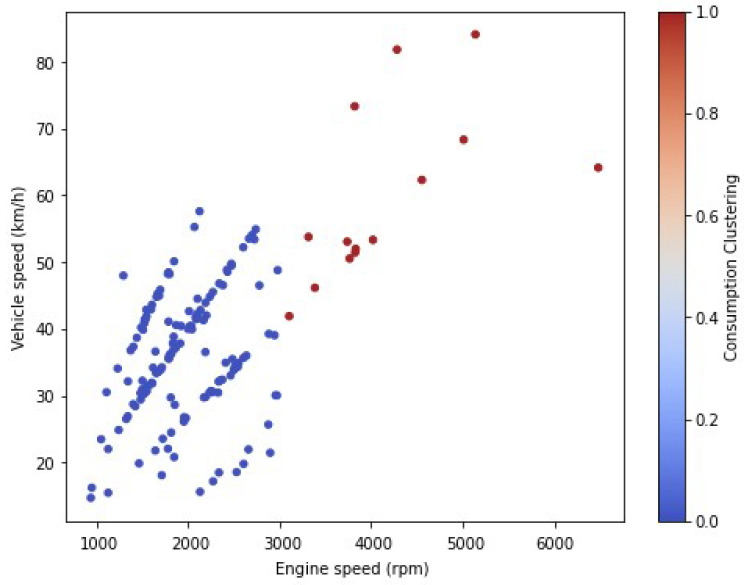
Consumption Clustering: vehicle speed vs. engine speed.

**Figure 12 sensors-23-02319-f012:**
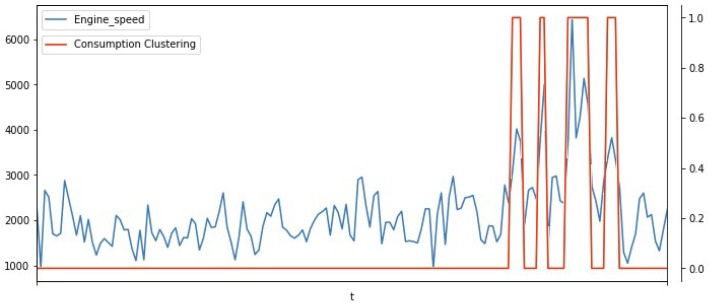
Engine speed (rpm) with consumption cluster (0–1).

**Figure 13 sensors-23-02319-f013:**
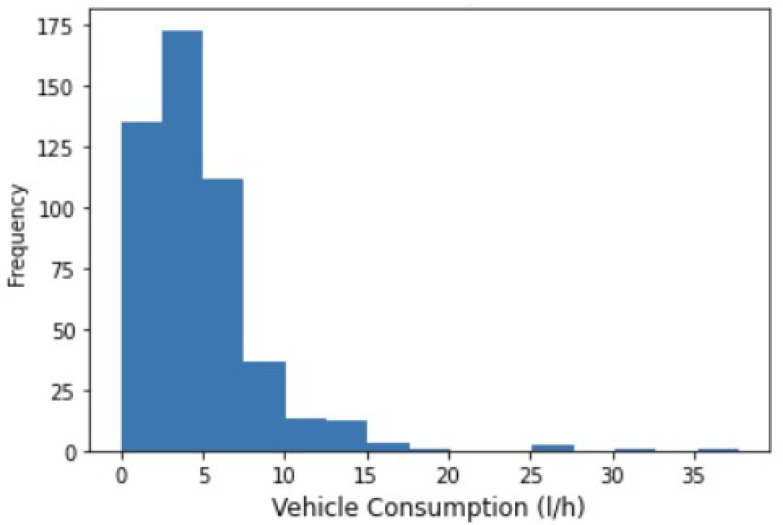
Vehicle consumption measured.

**Table 1 sensors-23-02319-t001:** Comparison of features of the commercial systems and IASE.

Features	ETAS	Vector	ATI	IASE
CCP	X	X	X	X
XCP	X	X	X	X
Data standardization (SI units)				X
Cloud integration				X
AI algorithms processing				X
Machine learning support				X
Save data to disk	X	X	X	X
Web interface for data visualization				X
Offline interface for data visualization	X	X	X	
Other communication protocols (e.g., FlexRay, LIN)	X	X	X	
JTAG	X	X	X	

**Table 2 sensors-23-02319-t002:** Experiment variables details.

Acquisition Period	Number of Variables
4 ms	28
5 ms	24
10 ms	60
100 ms	54
Sync with Cyl. 1	23
Sync with Cyl. 2	23
Sync with Cyl. 3	23
Sync with Cyl. 4	23

**Table 3 sensors-23-02319-t003:** Cost comparison between INCA and IASE.

INCA		IASE	
ES581 Connector	$1000.00	Processor Unit	$300.00
Computer	$2900.00	4G/GPS Modem	$55.00
License (per year)	$1600.00	Bluetooth Module	$4.50
		PCB	$30.00
		Other components	$40.00
Total	$5500.00	Total	$429.50

## Data Availability

Not applicable.

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
