# Peer review of "Low-Cost Data Acquisition System for Automotive Electronic Control Units"

_sensors, 2023, doi:10.3390/s23042319_

Round 1

Reviewer 1 Report

1. Capitalize the word "cost" in the title

2. In the abstract, the authors mentioned, "but at a lower cost (up to 13 times cheaper) and with more advanced features" what unit is used to measure the cost?

3. You should write the full name of the acronym, such as "ETAS."

4. I encourage the authors to use up-to-date references as much as possible, and my advice is to use references from the last five years.

5. How do the authors validate the AI algorithm? 

6. Why do the authors use such unsupervised learning? Is it possible to use supervised learning? Why? Why not?

7. Retitle the Conclusion as Conclusion and Recommendation.

Author Response

We appreciate your review and contributions.

Reviewer 2 Report

This paper has a very complete vehicle automatic data acquisition and transmission architecture and verificationThere are also functional comparisons of existing related systems. The system is quite complete.

Author Response

We appreciate your review.

Reviewer 3 Report

Considering the promising results, I would recommend expanding the conclusions, indicating more clearly the directions of applicability. I also recommend that more references be made to the financial implications on the automotive field.

Author Response

(The authors gave the same response as above.)

Reviewer 4 Report

Interesting paper in a state-of-the-art topic under the framework of Industry 4.0. You may find below some recommendations raised by the reviewer:

1) Indeed automotive manufacturing and electric vehicle manufacturing constitute a large percentage of the total market share. It is recommended to present some statistics and economic data to highlight the importance of the filed. You may check below some useful articles/reports:

a) https://www.statista.com/statistics/316786/global-market-share-of-the-leading-automakers/

2) You could also refer to Big Data analytics to electronics manufacturing due to the large amount of big data sets. 

a)  Li, B., Kisacikoglu, M.C., Liu, C., Singh, N. and Erol-Kantarci, M., 2017. Big data analytics for electric vehicle integration in green smart cities. IEEE Communications Magazine55(11), pp.19-25.

b) Mourtzis, D., Vlachou, E. and Milas, N.J.P.C., 2016. Industrial big data as a result of IoT adoption in manufacturing. Procedia cirp55, pp.290-295.

3) The authors state in Introduction (Contribution of this paper) that "they provide real-time communication through 4G/LTE network.". Actual real-time communication cannot be achieved with 4G/LTE networks due to latency and low bandwidth capabilities. Please check the following paper for more details regarding the implementation of 5G networks in future work:

a) Mourtzis D, Angelopoulos J, Panopoulos N. Smart Manufacturing and Tactile Internet Based on 5G in Industry 4.0: Challenges, Applications and New Trends. Electronics. 2021; 10(24):3175. https://doi.org/10.3390/electronics10243175

4) Use instead of "Background" the keyword "Communication protocols" in the title of Section 2. It reflects in a more clear manner the content of this section. 

5) Similar works in the field of Digital Manufacturing using IoT and Edge Computing for data acquisition and analysis can be found below. You may consider some of them towards the completeness of the literature review.

a) Mourtzis, D., Angelopoulos, J. and Panopoulos, N., 2022. Design and Development of an Edge-Computing Platform Towards 5G Technology Adoption for Improving Equipment Predictive Maintenance. Procedia Computer Science200, pp.611-619.

b)  Lydia Athanasopoulou, Harry Bikas, Alexios Papacharalampopoulos, Panos Stavropoulos & George Chryssolouris (2022) An industry 4.0 approach to electric vehicles, International Journal of Computer Integrated Manufacturing, DOI: 10.1080/0951192X.2022.2081363

6) Please explain in a more technical manner the term "Intelligent Acquisition and Analysis System". Which characteristics offer its intelligence? Is it adaptive? 

7) Please provide an Illustration of the System Architecture. For example, you should present the location of the sensors in the car, the communication with the server, the type of data and so on. 

8) Please ensure the high quality of the Figures. 

  1.  

Author Response

(The authors gave the same response as above.)

Round 2

Reviewer 1 Report

·        The authors mentioned that the accuracy of their method is close to 75% based on the precision metric, is this accuracy considered sufficient in this research? Why? If not, try to increase the accuracy of your model. Then, please compare your accuracy with other research in the same area.

Author Response

(The authors gave the same response as above.)
